# Emergent patterns of reef fish diversity correlate with coral assemblage shifts along the Great Barrier Reef

F. Javier González-Barrios [1] ✉, Sally A. Keith [1], Michael J. Emslie [2], Daniela M. Ceccarelli [2], Gareth J. Williams [3] & Nicholas A. J. Graham [1]

Escalating climate and anthropogenic disturbances draw into question how stable large-scale patterns in biological diversity are in the Anthropocene. Here, we analyse how patterns of reef fish diversity have changed from 1995 to 2022 by examining local diversity and species dissimilarity along a large latitudinal gradient of the Great Barrier Reef and to what extent this correlates with changes in coral cover and coral composition. We find that reef fish species richness followed the expected latitudinal diversity pattern (i.e., greater species richness toward lower latitudes), yet has undergone significant change across space and time. We find declines in species richness at lower latitudes in recent periods but high variability at higher latitudes. Reef fish turnover continuously increased over time at all latitudes and did not show evidence of a return. Altered diversity patterns are characterised by heterogeneous changes in reef fish trophic groups across the latitudinal gradient. Shifts in coral composition correlate more strongly with reef fish diversity changes than fluctuations in coral cover. Our findings provide insight into the extent to which classic macroecological patterns are maintained in the Anthropocene, ultimately questioning whether these patterns are decoupling from their original underlying drivers.

Quantification of biological diversity across large spatial scales has led to the formulation of general principles governing the organisation of life on Earth[1]. For instance, the latitudinal diversity gradient (LDG) where species diversity decreases from the tropics toward higher latitudes is one of the most documented biodiversity patterns on the planet[2,3]. However, recent exponential increases in human activities and recurrent climate extremes are hypothesised to have weakened the coupling of large-scale ecological patterns in communities with underlying natural processes; effectively, anthropogenic processes are emerging as novel driver[4,5]. For instance, marine species richness has declined at the equator due to anthropogenic global warming[6], and *Anolis* lizard distributions across the Caribbean are better predicted by shipping intensity than by geographical isolation as predicted by island biogeography theory[7]. Elsewhere, local human impacts have altered coral reef fish biomass and trophic structure to the extent that patterns depart from expectations across depth gradients[8]. While evidence is increasing indicating emergent changes in biodiversity patterns, the mechanisms and consequences of these changes across large spatial extents remain largely unknown.

Widespread species replacement is likely to be a ubiquitous pattern at a global scale and across biomes[9]. This reshuffling can be explained by changes in species composition as species gains and losses within ecological assemblages balance out[10]. Balancing effects can hide changes in common metrics often suggesting no impact[11,12] (e.g., species richness). Analysing species turnover therefore presents an ideal opportunity to reveal ecological disruptions even when richness remains stable. For instance, coral reef ecosystems have experienced widespread ecological changes over recent decades[13], and in

[1]Lancaster Environment Centre, Lancaster University, Lancaster, UK. [2]Australian Institute of Marine Science, Townsville, QLD, Australia. [3]School of Ocean Sciences, Bangor University, Menai Bridge, Anglesey, UK. ✉e-mail: j.gonzalezbarrios@lancaster.ac.uk

some cases, have shifted to new ecological states dominated by non-framework building groups[14]. As such, the reshuffling of species can affect the functioning of ecosystems[15]. Importantly, coral reefs are also capable of regaining their coral cover when stressful conditions are reduced[16], yet this recovery does not guarantee a reassembly of communities[17]. While associations between coral cover and fish assemblages might be weak, especially for some groups[18], the processes underlying how changes in benthic communities influence faunal groups such as reef fish composition across large spatial areas and temporal extents remain unclear. Indeed, new coral assemblage configurations might be expected to have consequences for the functional properties of their associated fauna. Understanding these consequences is crucial for predicting the broader ecological implications of widespread species replacement.

Climate change-induced thermal stress has severely impacted the Great Barrier Reef (GBR) in Australia, with coral bleaching and associated mortality becoming a major driver of the reef ecosystem state in recent decades[19]. Between 1998 and 2022, the GBR has been impacted six times by extreme warming events, with four of these episodes occurring between 2016 and 2022, resulting in 98% of the reefs being bleached at least once over that time[20]. For example, in 2016 the northern GBR was affected by a mass bleaching event resulting in a 30% loss of coral cover and shifts in coral composition[21]. However, anthropogenic warming is not the only threat affecting the GBR; disturbances such as cyclones[22], crown-of-thorns starfish outbreaks[23], declines in water quality[24] and fishing pressure[25] have widely altered the condition of its reefs. Temporal data are essential to reveal how large-scale diversity patterns respond to environmental fluctuations operating at ecological time scales[26,27]. With a large latitudinal gradient in species composition and environmental conditions[28], the GBR therefore offers an excellent system with which to explore how the long-term effect of climate change has affected coral communities across large spatial extents and whether this is shaping reef fish assemblages in the Anthropocene.

Here, we use the GBR Long-Term Monitoring Program (LTMP) of the Australian Institute of Marine Science, one of the most comprehensive monitoring programmes of coral reef ecosystems worldwide[29]. This dataset provides the opportunity to evaluate how reef fish assemblages have changed over multiple decades (1995–2022) and across a large latitudinal extent (>1200 km; 14°S to 24°S). We analyse how patterns of reef fish diversity have changed by examining local diversity ($\alpha$ diversity) and species dissimilarity ($\beta$ diversity). Specifically, we ask: (1) Have latitudinal patterns in reef fish species richness changed over time?; (2) Are these changes associated with turnover in community compositions of reef fish assemblages?; (3) How have these changes impacted the trophic group structure of reef fishes?; and (4) To what extent do fluctuations in coral cover and shifts of coral assemblages correlate with reef fish diversity patterns across large spatial extents? While there are many drivers of reef fish assemblages, we focused on contrasting the influence of coral cover versus coral composition as these are the key ecosystem engineers of the habitats on which reef fish depend, and many other potential drivers influence reef fish indirectly by impacting coral communities[30–32]. More broadly, our work provides insight into the extent to which classic macroecological patterns are changing in the Anthropocene.

In this work, we find that latitudinal gradients of reef fish diversity along the Great Barrier Reef (GBR) have shifted between 1995 and 2022. Our analysis shows a decline in species richness at lower latitudes, with increased variability at higher latitudes. Additionally, reef fish species dissimilarity has systematically increased over time, indicating persistent deviation from initial conditions. This biodiversity reshuffling has functional impacts on trophic groups, with reductions in omnivores, planktivores, and herbivores in the northern GBR, and increases in the southern regions. These changes are more closely associated with shifts in coral community composition than

fluctuations in coral cover. Our findings highlight ongoing alterations in $\alpha$ and $\beta$ diversity in the Anthropocene, providing evidence related to the mechanistic processes driving macroecological patterns changes over ecological time scales.

## Results

### Latitudinal diversity gradient changes in the Great Barrier Reef

Reef fish species richness followed the expected LDG (i.e., greater species richness toward lower latitudes), yet has undergone significant changes across space and time (Fig. 1b, Supplementary Fig. 1). During the period from 1995 to 2000, there was a linear relationship between latitude and diversity, with the lowest species richness recorded at higher latitudes. The period 2001–2005 saw the lowest species richness recorded in the central GBR with the highest species richness at higher latitudes. From 2011 to 2015 the lowest species richness was recorded at higher latitudes, while from 2016 to 2022 the relationship was linear and similar to the initial period (1995–2000). However, there was a significant change in species richness from the period 1995–2000 to the 2016–2022 one that varied with latitude; a decline in fish species richness at lower latitudes (mean change of species richness = −3.15, 95% CI = −4.25, −2.05; Fig. 1c–f) and an increase at higher latitudes (mean change of species richness = 3.69, 95% CI = 2.41, 4.9). The high temporal variability of species richness at higher latitudes is largely driven by the group of reefs in Capricorn-Bunker, the most southerly sector of the GBR. A sensitivity analysis that excluded the Capricorn-Bunker sector (Supplementary Fig. 2) confirms that the high variability in high-latitude species richness through time is limited to this sector and reveals a weaker change in the remaining sectors. It highlights that the strong change in species richness at high latitudes in the 2016–2022 period is due to an increase of species richness in the Capricorn-Bunker sector, whereas the decline in richness at low latitudes persists.

### Reef fish turnover increases along the Great Barrier Reef

To examine whether the recorded latitudinal changes in species richness were associated with species turnover, we examined the dissimilarity of reef fish assemblages using (1) a moving window pairwise comparison to analyse differences between years (year-to-year), and (2) a comparison of each year to the initial year of the time series of every reef. At the scale of the entire GBR, year-to-year fish species dissimilarity has increased progressively over time (Fig. 2a). For instance, the fish assemblage in the period from 2016 to 2022 had the highest dissimilarity compared to all other time intervals (Fig. 2b; mean = 0.305, 95% CIs = 0.292, 0.318) whereas the first two periods (1995–2000 and 2001–2005) had the lowest species dissimilarity (mean = 0.241, 95% CIs = 0.228, 0.255; mean = 0.234, 95% CIs = 0.220, 0.247; respectively). To examine whether the change from annual to biennial sampling influenced the results, we conducted a sensitivity analysis that considered reefs that were only surveyed biennially after 2006 ($n = 82$). These results also showed that the period from 2016 to 2022 had the greatest species dissimilarity (Supplementary Fig. 3a). The initial-year comparisons also revealed an increase in species dissimilarity over time without indication of a return (Supplementary Fig. 4a). For instance, the highest species dissimilarity occurred in the 2016–2022 period (Supplementary Fig. 4b; mean = 0.441, 95% CIs = 0.402, 0.480) whereas the lowest species dissimilarity occurred in the period from 1995 to 2000 (mean = 0.331, 95% CIs = 0.293, 0.370).

At the scale of individual latitudinal sectors, species dissimilarity also increased over time in all sectors (Fig. 2c). The southernmost Capricorn-Bunker sector had the greatest year-to-year species dissimilarity (Fig. 2d, mean = 0.331, 95% CIs = 0.316, 0.346). This sector was also the most dynamic, with a drop from 1996 to 2005, then showing two peaks of species dissimilarity hereafter, one in 2011 and another in 2017 (Fig. 2c). Interestingly, the northernmost Cooktown/Lizard Island sector showed an increase in species dissimilarity after

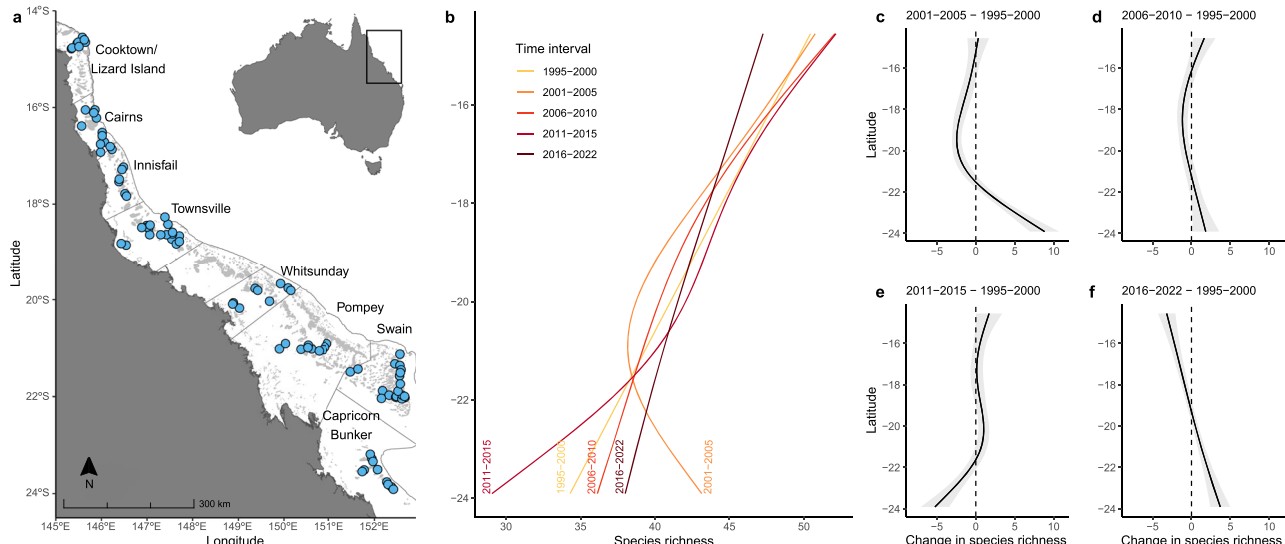

**Fig. 1 | Reef fish species richness (α diversity) across time and latitude. a** Blue circles on the map show reef sampling locations (*n* = 92 with three sites each per year) across the Great Barrier Reef. Borders in the map show the eight latitudinal sectors. **b** Species richness patterns across the latitudinal gradient based on predicted values in five-time intervals (1995–2000, 2001–2005, 2006–2010, 2011–2015 and 2016–2022) from the hierarchical generalised additive mixed model (HGAM). Each model line is shown in a gradient colour from yellow to red, representing the initial period to the more recent, respectively. Comparison of predicted values of the number of reef fish species with the initial time interval (i.e., 1995–2000) among all other subsequent periods are shown in **c** 2001–2005, **d** 2006–2010, **e** 2011–2015 and **f** 2016–2022. Grey bands in **c–f** represent the 95% CIs and dashed lines cross zero. Source data are provided as a Source Data file.

2010 followed by a drop and a new peak in 2016 before dropping again (Fig. 2d, mean = 0.230, 95% CIs = 0.204, 0.256). Species dissimilarity also increased over time across all latitudinal sectors using the initial-year approach (Supplementary Fig. 4c). We disentangled the Bray-Curtis dissimilarity (i.e., based on species abundances) among its balanced variation (i.e., turnover) and abundance gradient (i.e., nestedness) components. We found that the species turnover was the component that accounted for the increase in fish dissimilarity over time, while the nestedness component remained stable over time (Supplementary Fig. 5). These results were consistent with the Sørensen approach (i.e., presence-absence matrix) where we found that dissimilarity derived solely by turnover increased over time while nestedness did not show any pattern of change (Supplementary Fig. 6).

### Changes in the contribution of reef fish functional groups to community dissimilarity

At the GBR level, omnivores and planktivores were responsible for the greatest *net* negative contribution to community dissimilarity due to a strong decline of these groups (*net* = contribution to community dissimilarity across fish species within each functional group) (−15.73% and −15.7%, respectively). This equated to relative contributions (i.e., negative and positive contributions to community dissimilarity of each functional group relative to the rest) of −25.3% + 9.57% for omnivores and −32.2% + 16.5% for planktivores (Fig. 3a). Herbivores exhibited one of the greatest relative contributions to community dissimilarity (−24.2% + 24.7%) yet with a high *net* compensation between positive and negative values (*net* = 0.5%). The rest of the functional groups made smaller overall contributions to community change. However, we found that the contribution of different fish functional groups varies across the different latitudinal sectors (Fig. 3b, Supplementary Table 6). For instance, Capricorn-Bunker reefs had the greatest positive contribution of herbivores to community dissimilarity (*net* = 7%). When the Capricorn-Bunker sector was excluded from the analysis, we found that herbivores ranked among the main contributors to *net* negative contribution in the GBR, with omnivores and planktivores consistently identified as the main drivers of *net* negative community dissimilarity (Supplementary Fig. 7).

### Long-term coral cover and coral composition changes on the Great Barrier Reef

Over time, substantial changes in coral cover occurred across the GBR (Fig. 4a). The period with the greatest coral cover occurred during 1995–2000 (mean = 40.0%, 95% CIs = 36.1%, 43.8%; Fig. 4a, b) with a peak in 1999 (mean = 42.2%, 95% CIs = 37.6%, 46.8%). Subsequently, coral cover declined steadily, reaching its lowest period between 2011 and 2015 (mean = 31.5%, 95% CIs = 27.7%, 35.3%) with the lowest year in 2011 (mean = 31.1%, 95% CIs = 26.5%, 35.6%). A partial recovery was detected through the 2016–2022 period to an average of 36.0% (95% CIs = 32.2%, 39.8%), whereas 2022 was the year with higher coral cover levels of the period (mean = 36.9%, 95% CIs = 32.2%, 41.6%). Despite this recent recovery trend, the GBR did not regain coral cover levels observed at the beginning of the time series (but see estimations of coral cover using manta tow surveys[33]). Furthermore, we found a progressive increase in coral compositional dissimilarity over time (i.e., a rolling window comparing coral composition of most recently sampled years through time, Fig. 4c). The lowest compositional change occurred during the period 1995 to 2000 (mean = 0.220, 95% CIs = 0.185, 0.252; Fig. 4d). This was followed by a small increase in the period 2001–2005 (mean = 0.230, 95% CIs = 0.197, 0.263; Fig. 4d). The change in coral composition dissimilarity then began to increase more markedly after 2006, reaching its maximum levels in the periods 2011–2015 and 2016–2022 (mean = 0.270, 95% CIs = 0.237, 0.303 and mean = 0.267, 95% CIs = 0.234, 0.300, respectively; Fig. 4d).

Coral community composition across the GBR also shifted over time (ADONIS: *p*-value < 0.001; Fig. 4e). From 1995 to 2010, coral community composition was relatively stable and mainly characterised by *Acropora* branching-bottlebrush, *Acropora* digitate, *Acropora* tabulate-corymbose, *Merulina* and *Echinopora*. However, a greater shift in coral composition away from those dominant taxa started in 2011, whereby the ellipses depicting coral composition similarity (95% CIs) during the two most recent periods (2011–2015 and 2016–2022) do not overlap with those of any other time intervals (ADONIS: *p*-value < 0.001; Fig. 4e). Yet, the period 2016 to 2022 has seen a strong recovery of *Acropora* corals across much of the GBR (ADONIS: *p*-value < 0.001): the direction of the trend line from the period 2011–2015 to the 2016–2022 period

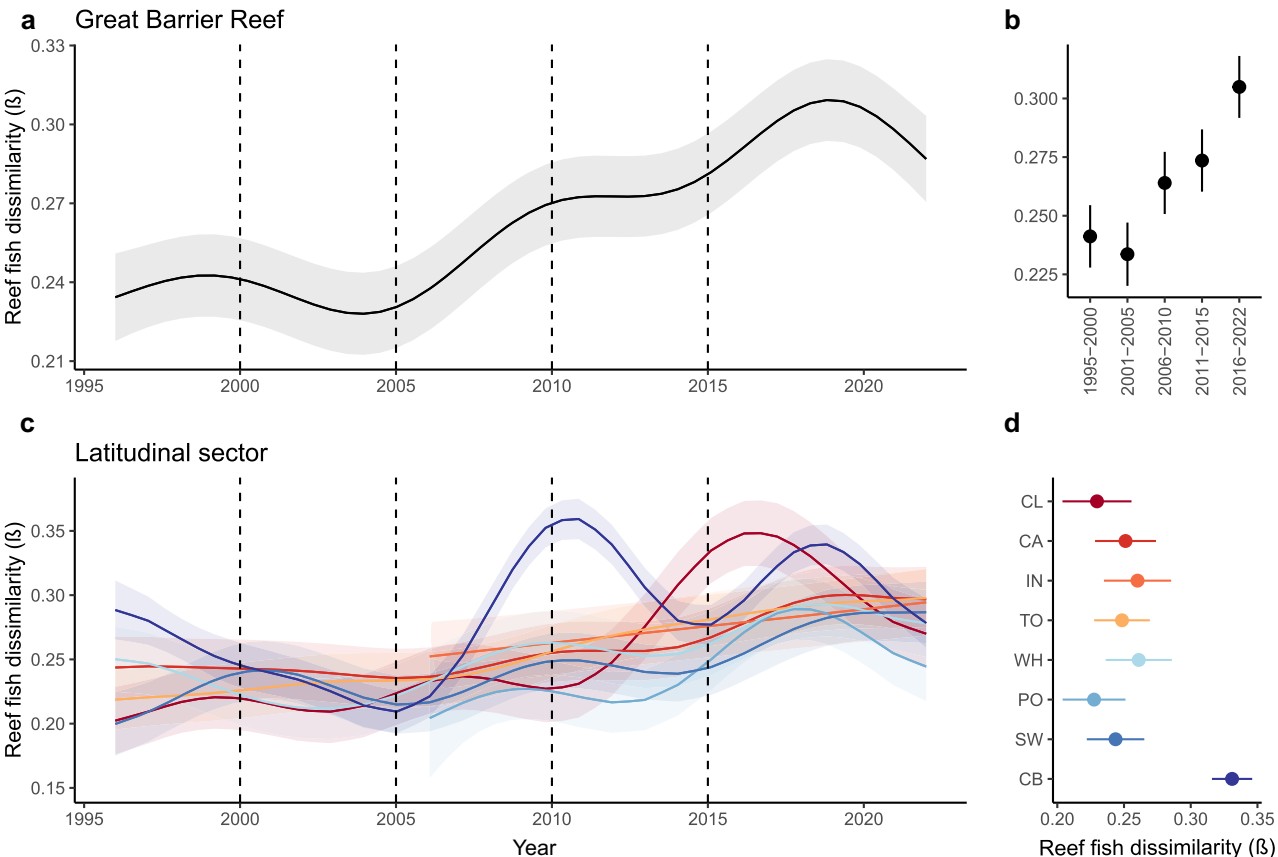

**Fig. 2 | Reef fish year-to-year dissimilarity (β diversity) across the Great Barrier Reef (GBR) and the latitudinal sectors.** Predicted values of the long-term changes in the reef fish turnover of 92 reefs with three sites each per year from hierarchical generalised additive mixed models (HGAM) according to the year-to-year dissimilarity across **a** the entire GBR and **c** individual latitudinal sectors. Shading bands in **a** and **c** represent 95% CIs. Predicted values from the generalised linear mixed model (GLMM) are shown in **b** for each time interval in the GBR. Number of reefs per period in **b**: 1995–2000 (*n* reefs = 46), 2001–2005 (*n* reefs = 46), 2006–2010 (*n* reefs = 92), 2011–2015 (*n* reefs = 92), and 2016–2022 (*n* reefs = 92). Vertical dashed lined in **a** and **c** represent time intervals depicted in **b**. Predicted values from the HGAM are shown in **d** for the whole temporal series of each latitudinal sector. Points in **b** and **d** represent the mean of the predicted values and lines represent the 95% CIs. Latitudinal sectors are arranged in descending order from north to south and in colours from red to blue, respectively. Codes and number of reefs sampled for each latitudinal sector are: Cooktown/Lizard Island (CL, *n* reefs = 8), Cairns (CA, *n* reefs = 11), Innisfail (IN, *n* reefs = 7), Townsville (TO, *n* reefs = 18), Whitsunday (WH, *n* reefs = 9), Pompey (PO, *n* reefs = 12), Swain (SW, *n* reefs = 17) and Capricorn-Bunker (CB, *n* reefs = 10). Source data are provided as a Source Data file.

aligns with the *Acropora* digitate (ACD) and *Acropora* tabulate & corymbose (ACTO) vectors (Fig. 4e).

### Correlation between reef fish diversity and coral cover/coral composition changes

Fish diversity change was more strongly correlated with the shift in coral composition (estimate = 0.188, 95% CIs = 0.151, 0.225) than it was with the rate of change of coral cover (estimate = 0.001, 95% CIs = <0.001, 0.002; Fig. 5). The shift in coral composition was three times more important a predictor than the change in coral cover (Supplementary Fig. 8). The effect of shifts in coral composition on fish dissimilarity was greatest in the Capricorn-Bunker and Cooktown/Lizard Island sectors and revealed steeper turnover patterns than the remaining sectors (estimate = 0.354, 95% CIs = 0.263, 0.444; estimate = 0.356, 95% CIs = 0.18, 0.21, respectively; Fig. 5d, e). Regardless of the weaker effect of the change in coral cover on reef fish turnover, there were some differences across latitudinal sectors (Supplementary Fig. 9). For instance, the greatest influence of change in coral cover on reef fish dissimilarity (estimate = 0.003, 95% CIs = 0.001, 0.006; Fig S9b) occurred in the Pompey sector, followed by Capricorn-Bunker and Cooktown/Lizard sectors (estimate = 0.003, 95% CIs = 0.001, 0.0037, estimate = 0.002, 95% CIs = 0.001, 0.005, respectively). Shifts in coral composition were found to have a negative influence on the

change in fish species richness, while the effect of the change in coral cover was not significant (estimate = −6.33, 95% CIs = −10.55, −2.11; estimate = 0.06, 95% CIs = −0.01, 0.13, respectively; Supplementary Fig. 10a–c). A greater negative effect of shifts in coral composition on the change of fish species richness was observed in the Capricorn-Bunker and Cairns sectors (Supplementary Fig. 10d, e), while no effect was detected in the remaining sectors.

## Discussion

Climate and anthropogenic pressures have intensified in recent decades[19,34], increasing uncertainty regarding the stability of large-scale biodiversity patterns. Here, we show that the shape of latitudinal gradients of reef fish diversity has changed along the GBR. Our analysis reveals declining fish species richness at lower latitudes and increasingly high variability of species richness at higher latitudes between 1995 and 2022. We also found a systematic increase in reef fish species dissimilarity over time, deviating from initial conditions with no evidence of a return. This reshuffling of diversity translates into functional consequences through gains or losses of different trophic groups. The northern section of the GBR exhibited a reduction in the abundance of omnivores, planktivores, and herbivores, while these groups increased in the southernmost sectors. Changes in reef fish diversity have a stronger correlation with changes in coral community composition,

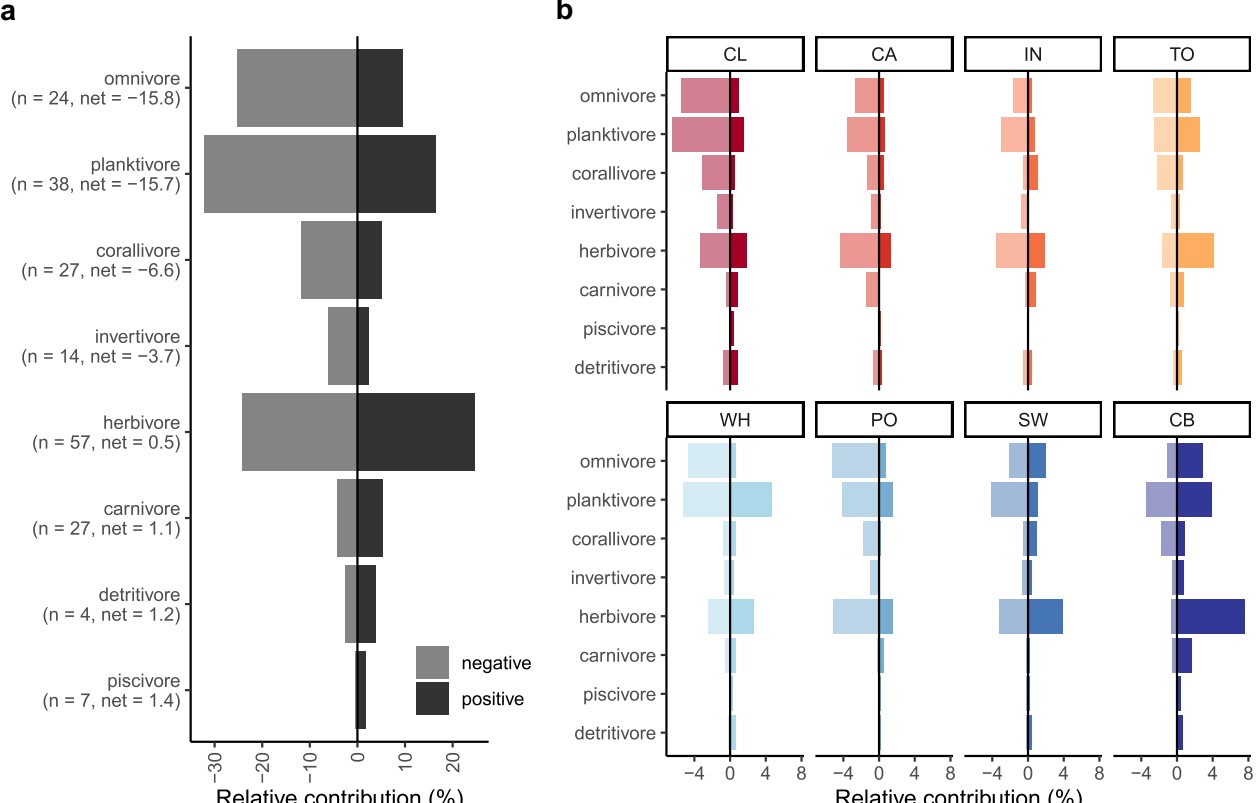

**Fig. 3 | Contribution of reef fish functional groups to community dissimilarity between the initial and last period in the Great Barrier Reef (GBR) and the latitudinal sectors.** Values are based on similarity percentage (SIMPER) analysis at the scales of **a** the GBR and **b** individual latitudinal sectors. Bars are species-level contributions summed by functional groups for species that increased (dark) or decreased (light) in abundance between the 1995–2000 and the 2016–2022 periods. The initial period of Innisfail (IN) and Pompey (PO) is the 2006–2010. Trophic groups are ordered in descending order according to the greatest net negative contribution (%) in **a**. Number of species (*n* =) and the net contribution to community dissimilarity (*net* =) for each functional group are shown in brackets. Relative contributions in **b** are according to overall values in **a**. Codes in **b** for latitudinal sectors are Cooktown/Lizard Island (CL), Cairns (CA), Innisfail (IN), Townsville (TO), Whitsunday (WH), Pompey (PO), Swain (SW) and Capricorn-Bunker (CB). Latitudinal sectors are ordered from lower (red) to higher latitudes (blue). Source data are provided as a Source Data file.

than overall coral cover. Collectively, we demonstrate that patterns in fish species richness (*α* diversity) and species turnover (*β* diversity) are changing and continuously moving forward in the Anthropocene.

Long-term impacts have shaped coral reef ecosystems in the GBR[35]. Yet, remarkable resilience in fish diversity has been observed at local scales[36,37]. Our study reveals that the well-recognised LDG, with diversity peaking near the tropics, has changed in shape through time. At low latitudes, the only period that showed a decrease of reef fish species richness was from 2016 to 2022, while the highly dynamic changes at higher latitudes were driven by the southernmost Capricorn-Bunker sector. Multiple disturbances have driven changes in the diversity patterns of the GBR. For instance, the period 2001–2005 witnessed the lowest species richness in the central GBR, likely associated with cyclone Tessi and a wave of crown-of-thorns starfish (CoTS) outbreaks[38,39]. Similarly, the 2011–2015 period showed the lowest species richness recorded at higher latitudes due to cyclones Hamish and Yasi and CoTS outbreaks[22,40,41]. Fluctuations of fish LDG have also been documented in the Northwest Atlantic continental shelf likely due to fluctuations in temperature[26,27]. At the global scale, recent studies show a decrease in species richness around the equator due to climate change[6,42]. In our study, the decline of species richness in the period 2016–2022 is likely associated with the increased frequency of cumulative impacts from acute disturbances including mass bleaching events, CoTS outbreaks, and tropical cyclones[20,33]. For decades, this latitudinal pattern of diversity was perceived as relatively

stable, yet, the unprecedented increase in anthropogenic disturbances such as climate change might have destabilised gradients in large-scale biogeographic patterns.

Species turnover means that changes in species composition do not necessarily equate to changes in species richness[11]. Shifts in species composition are reflected directly in *β* diversity approaches, making them potentially more sensitive indicators of community change than *α* diversity metrics. We found that an increase in reef fish dissimilarity (*β* diversity) has been a ubiquitous pattern along the latitudinal gradient of the GBR, with the Capricorn-Bunker sector showing more dynamic changes through time (Fig. 2c). Interestingly, this dissimilarity was mainly driven by species replacement (i.e., species turnover) rather than the loss of individuals from one year to the other without substitution (i.e., nestedness; Supplementary Figs. 5, 6). Increasing species turnover can be explained by different mechanisms. For example, unsteady levels of productivity and resource availability[43], a dynamic equilibrium) of species richness maintained by persistent colonisation and extinction (i.e., equilibrium model)[44] or species poleward distribution shifts in response to climate change[45]. On the GBR, variance in abundance among species is mainly because of persistent differences in the species' long-term abundances rather than their stochastic fluctuations[46] due to a high response diversity to environmental fluctuations[47]. Species turnover is one of the more widespread patterns of biodiversity change globally[9] with higher rates at the equator[48]. However, we show that trends in species turnover

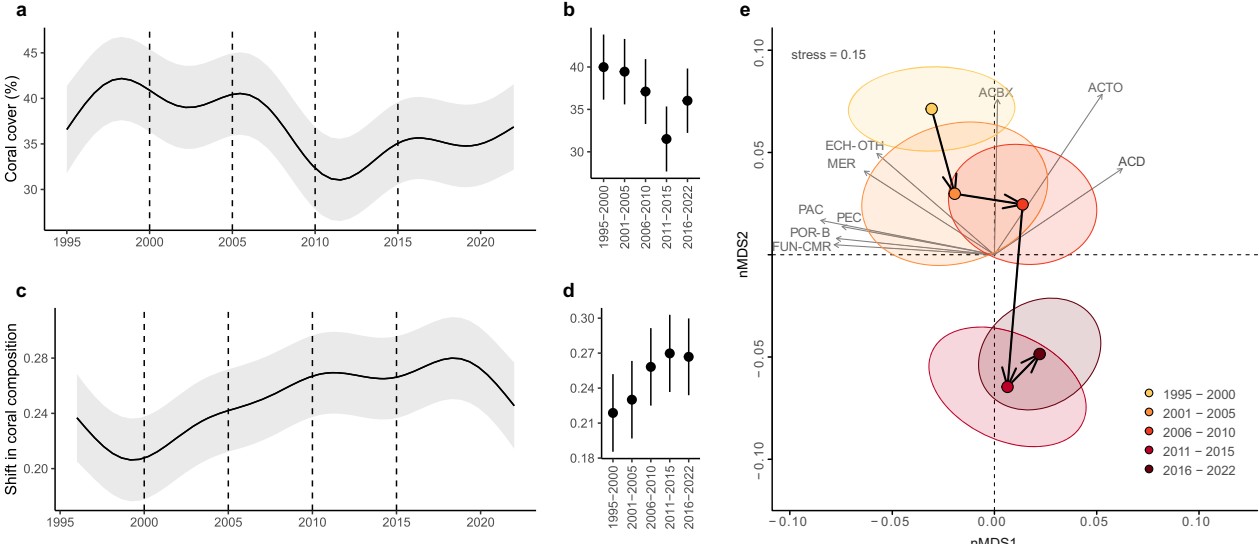

**Fig. 4 | Long-term changes of coral cover (%) and shift in coral composition on the Great Barrier Reef. a, b** Predicted values of the long-term changes of 92 reefs (three sites each per year) in coral cover and **c, d** shift in coral composition from hierarchical generalised additive mixed models (HGAM) in the Great Barrier Reef (GBR). **a, c** Solid black lines represent the mean of the fitted values from the HGAM and the 95% CIs are shown in grey bands. Predicted values from the generalised linear mixed model (GLMM) are the average values from each period for **b** coral cover (%) and **d** the shift in coral composition ($\beta$ diversity). **b, d** Number of reefs per period: 1995–2000 (n reefs = 46), 2001–2005 (n reefs = 46), 2006–2010 (n reefs = 92), 2011–2015 (n reefs = 92), and 2016–2022 (n reefs = 92). Vertical dashed lined in **a** and **c** represent time intervals in **b** and **d**. **b, d** Dots represent the mean and lines represent the 95% CIs. **e** Non-metric multidimensional scaling (nMDS), where dots represent the

weighted average (group centroid) of the coral composition of the reef sites in different periods. Ellipses represent the 95% CIs around the scores of each time interval. The permutational multivariate analysis of variance (ADONIS) showed significant differences in coral composition among the different time intervals ($p$-value < 0.05). Vector lines (grey arrows) show those corals that correlated >40% with the first two ordination axes; the length of the vector is proportional to the strength of the correlation and the direction indicates the relationship of each coral to the time interval groupings in multivariate space. In **e** codes for corals in grey are: *Acropora* branching & bottlebrush (ACBX), *Acropora* digitate (ACD), *Acropora* tabulate & corymbose (ACTO), *Echinopora* other (ECH-OTH), *Merulina* (MER), *Pachyseris* (PAC), *Pectinia* (PEC), *Porites* branching (POR-B), *Fungiidae* free-living (FUN-CMR). Source data are provided as a Source Data file.

have not only been maintained over the last three decades but are also accelerating in recent years, without any sign of stabilisation (Supplementary Fig. 3). We show that climate change is not only altering ecological assemblages from reference conditions[9] throughout large continuous natural systems such as the GBR. We demonstrate that these trends in species turnover have also substantially altered the latitudinal diversity patterns in the GBR, raising questions about potential consequences on the functioning of this ecosystem.

Shifts in coral communities can favour some species over others, and species' dependence on specific habitats determines their vulnerability[49]. We observed changes in fish communities due to species turnover with heterogeneous impacts on different functional groups along the GBR. Notably, declines in omnivores, planktivores and herbivores in the north and central GBR likely reflect changes in benthic communities. In contrast, the increase of these trophic groups in the southernmost Capricorn-Bunker sector of the GBR is likely to be due to a boom and bust of fast-growing *Acropora* due to storm impacts and subsequent coral recovery and reassembly. This has resulted in periods of very low coral cover, low habitat complexity and high algal productivity followed by high coral cover and low algae[33,50]. An increase in coral cover in the GBR along with a reduction in algae is likely to reflect reductions of omnivores and herbivores[51]. These cycles of impact and recovery have determined changes in reef fish $\alpha$ diversity in each period compared to the 1995-2000 period with reef fish responses varying within and among trophic groups in the GBR. Persistent stress due to climate change and exploitation results in changes in fish assemblages[52,53] and might jeopardise key functions and services provided by fish in the long term[54].

Climate extremes are escalating in the Anthropocene, resulting in negative outcomes for coral reef ecosystems[19,55]. Indeed, it is likely that cumulative stress is influencing how coral reefs respond to future

disturbances[20,56]. Concurrent with a rise in the frequency and intensity of climate-driven disturbances in the GBR[20,33], we observed a heightened shift in coral composition and turnover of fish assemblages between the 2011 and 2022 period. Nonetheless, the GBR has proved to be a resilient system, with many reefs recently recovering to or surpassing pre-disturbance coral cover[33]. However, recovery of coral cover can mask the influence of underlying compositional shifts in coral assemblages. For example, replacement coral species might not provide the same level of habitat complexity as the taxa they replaced[57] and in many cases result in negative outcomes for the associated fauna. Disruptions in crucial coral-fish relationships due to coral composition shifts can trigger cascading effects, leading to declines in fish abundance and diversity, as well as alterations in trophic structures and behaviour[32,58,59]. For example, changes in the dominance by *Pocillopora* rather than *Acropora* are likely to result in lower species richness and functional diversity of reef fishes[32] and might reflect bottom-up effects due to lower structural complexity[60]. Here, we show that coral composition plays an important role in the functional integrity of reef fish communities, contributing to distinct fish assemblages and trophic structures (Fig. 3). For instance, habitat degradation and repeated heatwaves have been characterised by a decline in specialist populations and declines of large-bodied warm-temperate vertebrates while broadly favouring generalist species[49,61,62]. Therefore, global studies have shown an increase in the occupancy of large-ranged species while small-ranged species have decreased[63]. While returning total coral cover may suggest a seemingly healthy reef, our findings highlight the importance of considering coral composition when assessing reef fish diversity change and alterations to trophic structures.

Classic and long-standing theories of biodiversity may exhibit reduced predictive power in a human-dominated world, requiring their re-evaluation to understand emergent ecosystem properties,

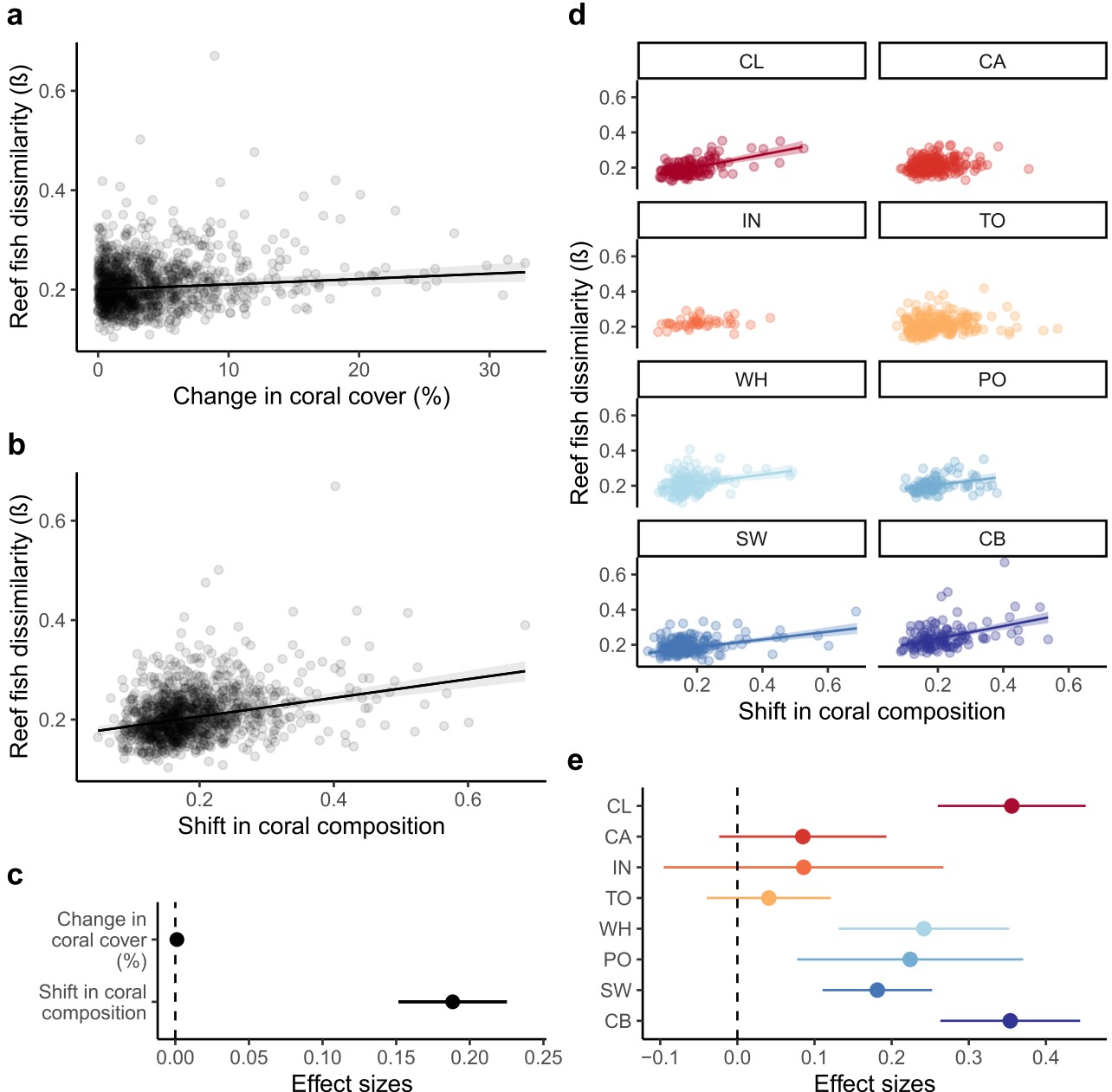

**Fig. 5 | Reef fish dissimilarity correlates more strongly with shifts in coral composition than changes in coral cover (%).** Predicted values of reef fish dissimilarity ($\beta$ diversity) of 92 reefs (three sites per reef per year) for **a** change in coral cover (%) and **b** shift in coral composition. Dots in **a** and **b** represent sites, and lines represent the fitted generalised linear mixed models (GLMM) (*n* reefs = 92). '*Reef*' and '*sShelf position*' were used as random effects in our models. Points in **c** represent means of the effect sizes of the GLMM and lines the 95% CIs (*n* reefs = 92). **d** Predicted values of reef fish dissimilarity with shifts in coral composition at latitudinal sector level: CL (*n* reefs = 8), CA (*n* reefs = 11), IN (*n* reefs = 7), TO (*n* reefs = 18), WH (*n* reefs = 9), PO (*n* reefs = 12), SW (*n* reefs = 17), CB (*n* reefs = 10).

Shading bands in **a**, **b** and **d** represent the 95% CIs. In (**c**) and (**e**), slopes are significant if their 95% CIs do not overlap the vertical dashed line at zero. Fitted lines for CA, IN, and TO in (**d**) are omitted as their effects are non-significant. Points in **e** represent the mean effect size of shifts in coral composition on reef fish dissimilarity for each latitudinal sector from the GLMM; lines represent 95% CIs. Codes in **d** and **e** are latitudinal sectors: Cooktown/Lizard Island (CL), Cairns (CA), Innisfail (IN), Townsville (TO), Whitsunday (WH), Pompey (PO), Swain (SW) and Capricorn-Bunker (CB) ordered in descending from lower to higher latitudes and colours from red to blue, respectively. Source data are provided as a Source Data file.

emergent dynamics of macroecological patterns and their underlying mechanisms[8,27,64]. Our study shows that the latitudinal gradient of reef fish diversity on the GBR has changed in shape over time and that increased reef fish species dissimilarity and decreasing fish species richness correlates more strongly with changes in coral composition, but to a lesser extent, also with changes in coral cover. It might be difficult to judge at this stage whether a weakening of the LDG is happening in the GBR due to climate change, however, future research will help to answer this question. However, coral reefs are not the only

ecosystems exhibiting disruptions from the predictions of classical theories. For example, invasive species have shifted behavioural traits and reduced variability, challenging predictions of the selective filter hypothesis[65]. Similarly, island biogeography theory reflects anthropogenic, rather than purely geographic, processes when human economic activities disrupt the distribution-isolation dynamics[7]. This increasing body of evidence suggests that ecological and biogeographic processes might have become disrupted in recent decades, with human socioeconomic and cultural activities becoming the

dominant drivers of changes in biodiversity patterns[64]. Further research should consider how human impacts are shaping large-scale biodiversity patterns at the heart of macroecology theory[66]. Our study provides insights towards understanding emergent properties of biodiversity and the direction natural systems are taking due to rapid environmental changes in the Anthropocene.

## Methods

### Study sites and data collection

We used coral and reef fish assemblage data from the Australian Institute of Marine Science's (AIMS) LTMP. The LTMP has monitored 92 reefs over 28 years (1995–2022), across eight latitudinal sectors (Cooktown/Lizard Island, Cairns, Innisfail, Townsville, Whitsunday, Pompey, Swain, and Capricorn-Bunker) spanning >1200 km (from 14°S to 24°S) of the GBR (Fig. 1a). Between 1995 and 2005, 46 reefs were monitored annually, then biennially thereafter. Then, from 2006 to 2020, an additional 46 reefs were surveyed biennially, as well as ten of the original 46 which continued to be surveyed annually. Six of the eight sectors have included survey reefs throughout the whole time series, while two sectors (Innisfail, Pompey) were added in 2006 (Supplementary Table 1).

Surveys were structured hierarchically. Within each reef, three sites in a standardised reef slope habitat were surveyed between 6 and 9 m depth. At each site, five permanent transects 5 m × 50 m wide were used to survey the abundance of large diurnal non-cryptobenthic reef-associated fishes from nine families covering a total of 198 reef fish species. Damselfishes (47 species) were counted separately using 1 m × 50 m wide belts along the same transects[67]. Photo transects were surveyed concurrently on the same transects to catalogue benthic assemblages, by taking a digital image every 1 m along the 50 m transects (50 images per transect), from which forty were randomly selected. The percentage cover of benthic groups was estimated from images using five points per image ([68]; $n = 200$) points per transect) and organisms were identified to the finest taxonomic resolution possible (usually genera) and placed into taxa/morphological groups (Supplementary Table 2).

### Changes in the diversity patterns of fish and coral assemblages

Long-term changes in the diversity of fish were investigated across a latitudinal gradient by quantifying changes in species richness ($\alpha$ diversity) and species dissimilarity (temporal $\beta$ diversity). Species abundances were pooled within each year and then used to calculate alpha diversity and temporal ß diversity indices per site. First, we assessed changes in $\alpha$ diversity of reef fish across the latitudinal gradient by pooling the total number of species at each site ($n = 5$ transects per year) for each survey year and in five-time intervals: 1995–2000, 2001–2005, 2006–2010, 2011–2015 and 2016–2022 (Supplementary Table 3). These periods ensured detailed spatial coverage without compromising temporal resolution. Notably, the earliest period (1995–2000) encompasses the first major mass bleaching event of 1998 recorded for the GBR. Furthermore, the 2016–2022 period had an unprecedented increase in the frequency of mass bleaching events[20,33]. Thus, we made a comparison of the species richness of fish between the initial period (i.e., 1995–2000) and then with all other subsequent periods.

Temporal $\beta$ diversity quantifies differences in the composition of a given assemblage between 2 (or more) years. We calculated temporal turnover in species composition (β) using the abundance-based Bray–Curtis dissimilarity metric[69] ($BC_d$). The data were square root transformed before the calculation of the $BC_d$ to homogenise variance. Values of $BC_d$ range from 0 to 1 which indicates perfect similarity or complete dissimilarity, respectively. Pairwise comparisons were calculated across survey years as:

$$BC_d = \frac{\sum \left( X_{ija} - X_{ijb} \right)}{\sum \left( X_{ija} + X_{ijb} \right)} \quad (1)$$

where $x_{ij}$ is the abundance of species $i$ on site $j$ between years $a$ and $b$. Temporal $\beta$ diversity can be estimated either 1) using a rolling window comparing most recently sampled years through time or 2) in comparison to a reference sample year. We first analysed temporal $\beta$ diversity changes using a moving window pairwise comparison to analyse differences between years. Pairwise comparisons were conducted year-to-year for reefs surveyed annually or every two years for reefs that were surveyed biennially. To control for potential sampling effort, a sensitivity analysis was conducted by subsetting all reefs that were biennially surveyed after 2006 ($n = 82$). Second, we analysed how reef fish assemblages have changed according to the initial sampling year of each reef and then progressively with all subsequent years. Furthermore, we disentangle the $BC_d$ among its balanced variation (i.e., turnover) and abundance gradient (i.e., nestedness) components[70]. We analysed temporal $\beta$ diversity under presence and absence data by using Sørensen beta diversity metric. Our Sørensen dissimilarity analysis was separated into two components accounting for the dissimilarity derived solely from turnover and the dissimilarity derived from nestedness[70].

### Statistical analysis

**Modelling patterns of reef fish diversity.** We generated hierarchical generalised additive mixed models (HGAMs) to analyse changes in $\alpha$ diversity of reef fish by fitting species richness at the site level ($n = 5$ transects per year) for each of the five temporal periods and across the latitudinal gradient (model 1; Supplementary Table 4). Given the high variation of species richness changes in the south, we conducted a sensitive analysis by excluding reef sites from Capricorn Bunker which is the southernmost sector of the GBR. Then, we fitted separate HGAMs to estimate changes for the two approaches of temporal $\beta$ diversity at the GBR level and for each of the eight latitudinal sectors, for the 'year-to-year' comparison (models 2 and 3) and the 'initial year' comparison (models 4 and 5). As a complementary analysis, we used General Linear Mixed Models (GLMM) to estimate the effect sizes for each time period (five periods) for the two approaches of temporal $\beta$ diversity: the year-to-year comparison (model 6) and the initial year comparison (model 7).

**Change in reef fish trophic groups and coral assemblages.** We grouped fish by the following dietary groups: omnivores, planktivores, herbivores, corallivores, invertivores, carnivores, detritivores and piscivores (Supplementary Table 5). We compared the contribution these groups made to differences in composition between the initial (1995–2000) and last period (2016–2022) in the GBR and among the latitudinal sectors. As surveys commenced in 2006 in the Innisfail and Pompey sectors, we considered 2006 to 2010 as the initial period for these sectors. We estimated the species contribution to compositional $BC_d$ (after square root transformation data) with a similarity percentage analysis (SIMPER)[71]. The percentage contribution to the SIMPER analysis was estimated as the absolute difference in abundance between the first and last time interval for each species $i$ at each site $j$. We present the percentage contribution of reef fishes within each functional group, summed across functional groups, and grouped by whether species contribute to positive or negative changes in abundance. The balance between the negative and positive contributions resulted in the *net* contribution to community dissimilarity (*net*). We also conducted a sensitive analysis to estimate overall contributions in the GBR by excluding reef sites in Capricorn–Bunker from the SIMPER analysis.

We assessed the changes in community composition in the GBR by estimating the coral composition in each of the five time periods. We performed a non-metric Multi-Dimensional Scaling (nMDS) based on the $BC_d$ of the square root transformed coral cover of the coral taxa. We estimate the weighted average centroids of the coral composition of the reef sites of each of the five time periods and ellipses representing the 95% CIs. We conducted a permutational multivariate

analysis of variance using distance matrices and assessed the sums of squares for each treatment to test differences in coral composition among the different time intervals.

**Benthic predictors of reef fish diversity patterns.** We first estimated temporal trends of coral cover (%) and shifts in coral composition (based on the Bray–Curtis dissimilarity metric, $BC_d$) at the site level ($n = 5$ transects per year) by using HGAM models (model 8 and 9, respectively). We also used general linear mixed models (GLMM) to estimate the average coral cover (%) and shift in coral composition of each time period (models 10 and 11, respectively). We also standardised both metrics to compare the relative magnitude of the effects on reef fish turnover ($\beta$). Furthermore, we analysed to what extent the change in coral cover and shifts in coral composition correlate with fish dissimilarity ($\beta$ diversity) and the change in species richness ($\alpha$ diversity) at the GBR (models 12 and 13) and for the latitudinal sectors (models 14, 15 and 16). We estimated the annual rate of change of (1) coral cover and (2) fish species richness as:

$$\frac{(V_f - V_i)}{t} \qquad (2)$$

where $V_i$ and $V_f$ are the initial and final coral cover and fish richness estimates, respectively, and $t$ is the number of years elapsed between surveys. We transformed the rate of change of coral cover into its absolute value to account for unidirectional changes. We estimated the year-to-year shift in coral composition ($\beta$) using the $BC_d$ built from square root transformed data.

**Model structures and assumptions.** Given the hierarchical structure of our data ($n = 98$ reefs) and to account for their variation within models we use '*reef*' ($n = 3$ sites) as a random effect for all our models. Therefore, we include cross-shelf position (i.e., inner, middle and outer reefs) as a random effect in our models since it has been recognised as an important source of variation in reef fish in the GBR[72,73]. To control for possible temporal autocorrelation in models 1 to 5 and 8 to 9, we added a correlation structure of the standard class autoregressive process of order 1 (corAR1), using '*year*' as the correlation variable. Model assumptions were validated with residual plots. For the HGAMs, we used four *knots* when modelling latitude (model 1) and eight *knots* to time (models 2–5 and 8–9), this allowed for potential variation in the smoother while keeping computation time low[74]. We used the *gam.check* and *k.check* functions to validate patterns in residual plots and determine whether *knots* used in our models were fitted appropriately.

$\beta$ metrics were estimated using *beta.pair.abund* function in the *betapart* package[75]. The permutational multivariate analysis was performed using the *adonis* function from the *vegan* package as well as the SIMPER and nMDS multivariate analyses[76]. HGAMs were generated using the *gam* function from the *mgcv* package[77] and GLMMs were generated with the *lmer* function from the *lme4* package[78]. All models, analyses and figures were generated in R version 4.3.0[79].

**Reporting summary**

Further information on research design is available in the Nature Portfolio Reporting Summary linked to this article.

## Data availability

All data used in this manuscript are publicly available upon request from the Australian government's Australian Institute of Marine Science: monitoring@aims.gov.au. Source data are provided with this paper.

## Code availability

We provide R code associated with this study at an open-source repository (https://doi.org/10.5281/zenodo.13963534).

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

## Acknowledgements

We thank James Robinson, Laura Richardson, Eva Maire, and three
anonymous referees for stimulating discussions that greatly improved
the manuscript. We acknowledge the Australian Institute of Marine
Science (AIMS) that primarily provides funding for the Long-Term
Monitoring Program. Additional funding was provided by the CRC Reef
Research Centre, the Australian Government's Marine and Tropical
Sciences Research Facility (MTSRF), the National Environmental
Research Program (NERP), and the National Environmental Science
Program (NESP). F.J.G.-B. was funded by a Natural Environment
Research Council (NERC) studentship with the Envision Doctoral
Training Partnership (NE/S007423/1).

## Author contributions

F.J.G.B., N.A.J.G., S.A.K. and G.J.W. conceived and designed the study;
M.J.E. and D.M.C. collected data; F.J.G.B. performed modelling, ana-
lysed output data and wrote the original draft of the manuscript, and all
authors contributed substantially to revisions.

## Competing interests

The authors declare no competing interest.

## Additional information

**Supplementary information** The online version contains
supplementary material available at

F. Javier González-Barrios.

**Peer review information** *Nature Communications* thanks Chhaya
Chaudhary and the other, anonymous, reviewer for their contribution to
the peer review of this work. A peer review file is available.

