## [Transparent Peer Review file · Nature Communications]

Emergent patterns of reef fish diversity correlate with coral assemblage shifts along the Great Barrier Reef

Corresponding Author: Mr Francisco Gonzalez Barrios

Version 0:

Reviewer comments:

Reviewer #2

(Remarks to the Author)

Based on my previous review, I see major changes have been applied to the manuscript. The current version is much more clear, concise and has all the adequate information. The new information on data added in supplementary tables S1 and S2 adds data transparency. The new analysis with Sorenson index makes the results robust. In my opinion, all the raised concerns are satisfied, and I recommend the publication of this important work.

Reviewer #3

(Remarks to the Author)

I reviewed a previous version of this ms, and I remain enthusiastic about it overall - there are interesting results about the temporal dynamics of a key macroecological pattern, the latitudinal diversity gradient, along the GBR. I will focus my comments here on the extent to which the authors have addressed concerns expressed in that original review. I'm pleased to see a comprehensive response, reflected in substantial changes to the manuscript, which have largely covered the points I made originally. I think this revision is a much better representation of what has actually been found, and avoids some of the over-interpretation - particularly with regards to putative causal relationships - that was present in the original submission. I think the main title maybe still over emphasises the correlation with coral assemblages, and lacks an explicit mention of the temporal dimension that is a strength of this work. I would favour something like: Temporal dynamics of emergent patterns in reef fish diversity along the Great Barrier Reef. But I would not insist on this, and retaining 'correlate with coral assemblage shifts' is fine.

I will briefly comment on responses to my main concerns, and then make some additional minor comments / requests / suggestions.

Systematic change since baseline versus annual variation - this has been really well addressed. I particularly appreciate Fig S1 which shows the site-level data at each latitude for each time period. It is quite striking that the truncation of the x axis in fig 1b (where only model fits are shown) does tend to emphasise differences that are much less apparent on fig S1 (where individual richness values are shown) - but I think providing both at least allows the reader to judge for themselves. I also think the inclusion of the nestedness/turnover analysis is a good addition. This is presented in quite a descriptive way with no predictions based on plausible scenarios of changes in species occurrence, but I think that it is adequate.

Correlation or causation? This has been well addressed and the language and framing of the results is now much more in keeping with what the data and analyses actually show. There are a couple of places where I think minor changes are still warranted:

L316 - "We show that climate change is not only altering ecological assemblages from reference conditions (Blowes et al. 2019) but occurring in large continuous natural systems such as the GBR" - you do not directly test for an impact of climate change, and you do not show that climate change is occurring in the GBR. Suggest rephrasing as something like: "Our results support the view that climate change can alter ecological assemblages from reference conditions (Blowes et al.

2019) throughout large continuous natural systems such as the GBR.”

L339 - “We show how the intensifying regime of a range of disturbances in the period 2011 to 2022, particularly the increase in the frequency of mass bleaching events (Emslie et al. 2024; Hughes et al. 2021), has increased the shift in coral composition and turnover of fish assemblages.” Again - I think this statement goes beyond what you have actually shown - nothing in your data or analysis directly addresses intensifying disturbances or increased frequency of coral bleaching. Suggest rephrasing.

The minor points that I raised have largely been addressed. One additional paper that might be helpful to reference, maybe in the concluding statements, is Kevin Gaston's 2004 perspective <https://doi.org/10.1016/j.baae.2004.05.001> on why we need to better incorporate human impacts into macroecology - I think this could help just emphasise the generality of these phenomena.

Some additional minor points that I noted on reading this revised version:

L111-112: what are the ‘means’ reported in this sentence - mean change in species numbers? It would be clearer to state this if so. Likewise I think the x axis labels in fig 1C would be better as ‘Change in fish species richness’

Fig 1B: could you label the lines directly with the time interval? e.g. at the bottom of each line? I think this would make it easier for the viewer to follow the trajectory for each time period. (This is not a requirement but rather something that could improve the figure if it is easy to do without adding clutter.)

Fig 2C - the shaded confidence interval for Pompey seems to start only in 2006, which is when the time series starts - but the fitted line seems to start in 1995 - is this correct?

L168 through to Fig 3 - can you explain this ‘net negative / positive’ contribution more clearly, as I found it confusing. Does a high net negative contribution to community dissimilarity mean that omnivore and planktivore communities tend to remain constant and similar (so they have a small effect on change in dissimilarity)? Or does it mean dissimilarity is more strongly affected by declines in omnivores and planktivores than by increases in these groups? In the methods (L455) this is explained as “We present the percentage contribution of reef fishes within each functional group, summed across functional groups, and grouped by whether species contribute to positive or negative changes in abundance” - which I think suggests the second explanation. But it would be useful to provide additional clarification, given that this is one of your key results.

L237 - this sentence is still ambiguous “The shift in coral composition had a stronger correlation with fish diversity change (mean = 0.188, 95% 238 CIs = 0.151, 0.225) than the rate of change of coral cover (mean = 0.001, 95% CIs = <0.001, 0.002; Fig. 5). This equated that the shift in coral composition was three times more important a predictor than the change in coral cover (Fig. S8)” Suggest changing to: “Fish diversity change was more strongly correlated with the shift in coral composition (mean = 0.188, 95% 238 CIs = 0.151, 0.225) than it was with the rate of change of coral cover (mean = 0.001, 95% CIs = <0.001, 0.002; Fig. 5). The shift in coral composition was three times more important a predictor than the change in coral cover (Fig. S8)” As mentioned above, I would recommend using a term more informative than just ‘mean’ in the parentheses too - ‘mean correlation’ here I think?

Response to reviewer comments

Reviewer #2 (Remarks to the Author):

Based on my previous review, I see major changes have been applied to the manuscript. The current version is much more clear, concise and has all the adequate information. The new information on data added in supplementary tables S1 and S2 adds data transparency. The new analysis with Sorenson index makes the results robust. In my opinion, all the raised concerns are satisfied, and I recommend the publication of this important work.

We appreciate all of your positive and constructive comments, which helped improve this work.

Reviewer #3 (Remarks to the Author):

I reviewed a previous version of this ms, and I remain enthusiastic about it overall - there are interesting results about the temporal dynamics of a key macroecological pattern, the latitudinal diversity gradient, along the GBR. I will focus my comments here on the extent to which the authors have addressed concerns expressed in that original review. I'm pleased to see a comprehensive response, reflected in substantial changes to the manuscript, which have largely covered the points I made originally. I think this revision is a much better representation of what has actually been found, and avoids some of the over-interpretation - particularly with regards to putative causal relationships - that was present in the original submission. I think the main title maybe still over emphasises the correlation with coral assemblages, and lacks an explicit mention of the temporal dimension that is a strength of this work. I would favour something like: Temporal dynamics of emergent patterns in reef fish diversity along the Great Barrier Reef. But I would not insist on this, and retaining 'correlate with coral assemblage shifts' is fine.

We are pleased to see that the changes we made were largely satisfactory regarding the comments you made on a previous version of this manuscript. We appreciate the suggestion of an alternative title for this work, however feel the current title highlights that our study not only evaluates changes in reef fish diversity patterns, but also the interesting association with coral assemblage shifts. We strongly feel this is a novel element of the contribution and so would favour retaining the existing title.

I will briefly comment on responses to my main concerns, and then make some additional minor comments / requests / suggestions.

Systematic change since baseline versus annual variation - this has been really well addressed. I particularly appreciate Fig S1 which shows the site-level data at each latitude for each time period. It is quite striking that the truncation of the x axis in fig 1b (where only model fits are shown) does tend to emphasise differences that are much less apparent on fig S1 (where individual richness values are shown) - but I think providing both at least allows the reader to judge for themselves. I also think the inclusion of the nestedness/turnover analysis is a good addition. This is presented in quite a descriptive way with no predictions based on plausible scenarios of changes in species occurrence, but I think that it is adequate.

Many thanks for this positive feedback

Correlation or causation? This has been well addressed and the language and framing of the results is now much more in keeping with what the data and analyses actually show. There are a couple of places where I think minor changes are still warranted:

L316 - "We show that climate change is not only altering ecological assemblages from reference conditions (Blowes et al. 2019) but occurring in large continuous natural systems such as the GBR" - you do not directly test for an impact of climate change, and you do not show that climate change is occurring in the GBR. Suggest rephrasing as something like: "Our results support the view that climate change can alter ecological assemblages from reference conditions (Blowes et al. 2019) throughout large continuous natural systems such as the GBR."

Changed as suggested.

L339 - "We show how the intensifying regime of a range of disturbances in the period 2011 to 2022, particularly the increase in the frequency of mass bleaching events (Emslie et al. 2024; Hughes et al. 2021), has increased the shift in coral composition and turnover of fish assemblages." Again - I think this statement goes beyond what you have actually shown - nothing in your data or analysis directly addresses intensifying disturbances or increased frequency of coral bleaching. Suggest rephrasing.

Thanks for your suggestion. We have changed the statement to: "Concurrent with a rise in the frequency and intensity of climate-driven disturbances in the GBR (Emslie et al. 2024; Hughes et al. 2021), we observed a heightened shift in coral composition and turnover of fish assemblages between the 2011 and 2022 period."

The minor points that I raised have largely been addressed. One additional paper that might be helpful to reference, maybe in the concluding statements, is Kevin Gaston's 2004 perspective <https://doi.org/10.1016/j.baae.2004.05.001> on why we need to better incorporate human impacts into macroecology - I think this could help just emphasise the generality of these phenomena.

Many thanks. We have added a new statement in the conclusion based on the Gaston 2004 paper.

Some additional minor points that I noted on reading this revised version:

L111-112: what are the 'means' reported in this sentence - mean change in species numbers? It would be clearer to state this if so. Likewise I think the x axis labels in fig 1C would be better as 'Change in fish species richness'

Done as suggested

Fig 1B: could you label the lines directly with the time interval? e.g. at the bottom of each line? I think this would make it easier for the viewer to follow the trajectory for each time period. (This is not a requirement but rather something that could improve the figure if it is easy to do without adding clutter.)

Thanks for your suggestion. We have added the time interval at the bottom of each line.

Fig 2C - the shaded confidence interval for Pompey seems to start only in 2006, which is when the time series starts - but the fitted line seems to start in 1995 - is this correct?

This has now been fixed. The correct one is from 2006 when the time series for that sector starts. The same was fixed for Innisfail.

L168 through to Fig 3 - can you explain this 'net negative / positive' contribution more clearly, as I found it confusing. Does a high net negative contribution to community dissimilarity mean that omnivore and planktivore communities tend to remain constant and similar (so they have a small effect on change in dissimilarity)? Or does it mean dissimilarity is more strongly affected by declines in omnivores and planktivores than by increases in these groups? In the methods (L455) this is explained as "We present the percentage contribution of reef fishes within each functional group, summed across functional groups, and grouped by whether species contribute to positive or negative changes in abundance" - which I think suggests the second explanation. But it would be useful to provide additional clarification, given that this is one of your key results.

Thanks for pointing out this statement which was indeed confusing. It refers to the second explanation you mention. We have now made it clear: "At the GBR level, omnivores and planktivores were responsible for the greatest net negative contribution to community dissimilarity due to a strong decline of these groups (net = balance between the negative and positive contribution to community dissimilarity across fish species within each functional group) to community dissimilarity (-15.73% and -15.7%, respectively)."

L237 - this sentence is still ambiguous "The shift in coral composition had a stronger correlation with fish diversity change (mean = 0.188, 95% 238 CIs = 0.151, 0.225) than the rate of change of coral cover (mean = 0.001, 95% CIs = <0.001, 0.002; Fig. 5). This equated that the shift in coral composition was three times more important a predictor than the change in coral cover (Fig. S8)" Suggest changing to: "Fish diversity change was more strongly correlated with the shift in coral composition (mean = 0.188, 95% 238 CIs = 0.151, 0.225) than it was with the rate of change of coral cover (mean = 0.001, 95% CIs = <0.001, 0.002; Fig. 5). The shift in coral composition was three times more important a predictor than the change in coral cover (Fig. S8)" As mentioned above, I would recommend using a term more informative than just 'mean' in the parentheses too - 'mean correlation' here I think?

Thanks. We have made the changes according to your suggestion.